# A Green Bioactive By-Product Almond Skin Functional Extract for Developing Nutraceutical Formulations with Potential Antimetabolic Activity

**DOI:** 10.3390/molecules28237913

**Published:** 2023-12-03

**Authors:** Patrizia Picerno, Lucia Crascì, Patrizia Iannece, Tiziana Esposito, Silvia Franceschelli, Michela Pecoraro, Virgilio Giannone, Anna Maria Panico, Rita Patrizia Aquino, Maria Rosaria Lauro

**Affiliations:** 1Department of Pharmacy, University of Salerno, Via Giovanni Paolo II, 84084 Fisciano, Italy; ppicerno@unisa.it (P.P.); tesposito@unisa.it (T.E.); sfranceschelli@unisa.it (S.F.); mipecoraro@unisa.it (M.P.); aquinorp@unisa.it (R.P.A.); 2Department of Drug and Health Sciences, University of Catania, Viale A. Doria, 95100 Catania, Italy; luciacrasci79@gmail.com (L.C.); panico@unict.it (A.M.P.); 3Department of Chemistry and Biology, University of Salerno, Via G. Paolo II 132, 84100 Fisciano, Italy; piannece@unisa.it; 4Unesco Chair Salerno, Plantae Medicinales Mediterraneae, University of Salerno, 84084 Fisciano, Italy; 5Department of Agricultural and Forest Sciences, University of Palermo, Viale delle Scienze Ed.4, 90128 Palermo, Italy; giannonevirgilio@gmail.com

**Keywords:** almond skins, Eudraguard^®^ Natural, DPPH, ABTS, ORAC test, MMP-9, MMP-2, AGEs, spray-drying technologies, MMT assay

## Abstract

(1) Background: almond peels are rich in polyphenols such as catechin and epicatechin, which are important anti-free-radical agents, anti-inflammatory compounds, and capable of breaking down cholesterol plaques. This work aims to evaluate the biological and technological activity of a “green” dry aqueous extract from Sicilian almond peels, a waste product of the food industry, and to develop healthy nutraceuticals with natural ingredients. Eudraguard^®^ Natural is a natural coating polymer chosen to develop atomized formulations that improve the technological properties of the extract. (2) Methods: the antioxidant and free radical scavenger activity of the extract was rated using different methods (DPPH assay, ABTS, ORAC, NO). The metalloproteinases of the extracts (MMP-2 and MMP-9), the enhanced inhibition of the final glycation products, and the effects of the compounds on cell viability were also tested. All pure materials and formulations were characterized using UV, HPLC, FTIR, DSC, and SEM methods. (3) Results: almond peel extract showed appreciable antioxidant and free radical activity with a stronger NO inhibition effect, strong activity on MMP-2, and good antiglycative effects. In light of this, a food supplement with added health value was formulated. Eudraguard^®^ Natural acted as a swelling substrate by improving extract solubility and dissolution/release (4) Conclusions: almond peel extract has significant antioxidant activity and MMP/AGE inhibition effects, resulting in an optimal candidate to formulate safe microsystems with potential antimetabolic activity. Eudraguard^®^ Natural is capable of obtaining spray-dried microsystems with an improvement in the extract‘s biological and technological characteristics. It also protects the dry extract from degradation and oxidation, prolonging the *shelf life* of the final product.

## 1. Introduction

Almonds (*Prunus dulcis*, Mill.) are among the richest natural food sources of vitamin E and polyphenols. In particular, almond skins are a potential source of bioactive components like vitamins, free amino acids, minerals, and polyphenols such as flavanols, primarily catechin and epicatechin (total monomer concentration of 7.8 mg/100 g) [1,2,3].

Once introduced to the body, catechins block free radicals, preventing DNA and protein damage. They also reduce inflammation, a triggering factor in metabolic syndrome, and decrease the degradation of cholesterol plaques, minimizing the incidence of cardiovascular diseases and tumors [4,5]. In the literature, some almond skin extracts showed activities similar to those of catechins. They have hepatoprotective properties due to inhibiting hepatocyte lipid peroxidation and cytotoxicity [6]. Furthermore, in a precedent work, an almond skin acetonic extract was studied to produce a cyclodextrin/extract complex. It showed to be active on TNF-α-inhibition, one of the leading mediators of intestinal inflammation [7]. These effects are mainly due to the polyphenol’s antioxidant and radical scavenging properties [8,9,10]. Indeed, reactive oxygen (ROS) and nitrogen (RNS) species are factors strictly related to inflammation. In the presence of oxidative stress, the glycation end products (AGEs) accumulated within body tissues contribute to raising the levels of pro-inflammatory factors, causing the upregulating of matrix metalloproteinases’ (MMPs’) production [9,11,12]. Polyphenols can act on various inflammatory disease factors, inhibiting different steps of the glycation process and the MMPs’ activities [9,13,14,15,16]. Polyphenols with multiple hydroxyl groups, such as catechins, can react with ROS and RNS in a termination reaction, breaking the new radical generation cycle [17] and capturing α-dicarbonyl species responsible for forming mono- and di-adducts. Thus, they inhibit the formation of AGEs [15]. In the present study, in the first step, a decoction green method was applied to obtain an aqueous almond skin extract (DSA) rich in polyphenols, mainly catechins. The factors contributing to the potential anti-inflammatory and dysmetabolic activity of almond skins were studied. The total polyphenol content (TPC; generally, 10-fold higher than that of the whole seed) [3,18], the antioxidant activity, the antiglycation effect, the metalloproteases’ (MMPs’) inhibition, and the safety of the extract were studied. In particular, different in vitro antioxidant approaches were applied, and MMP inhibition activity on MMP-2 and MMP-9 was tested. In a preliminary test, DPPH (stable 2,2-diphenyl-1-picrylhydrazyl radical) and ABTS (2,2′-Azino-bis(3-ethylbenzthiazoline-6-sulfonic acid) tests were carried out. The DPPH test is an economic method that requires no special equipment, used to estimate the antioxidant properties of food extracts [9,18]. Following the first screening, an ABTS assay was used to estimate the antioxidant activity of hydrophilic and lipophilic food substances. ABTS is correlated to the ORAC (Oxygen Radical Absorbance Capacity) USDA database of polyphenolics [19]. Therefore, the ORAC test, the “golden standard” of the food industry [19] developed by the NIA (National Institute on Aging) at the National Institutes of Health (NIH), was carried out. The antioxidant effect depends on the radical species involved, so, finally, the NO extract scavenger effect was evaluated. The advanced glycation end products (AGEs) formation was also investigated since it is strictly related to MMPs’ activity [20]. The compounds’ effect on cell viability was also tested. 

In light of these results, nutraceutical formulations were developed in a second step. Eudraguard^®^ Natural (EN), a gluten-free and certified GMO-free maize starch-based (starch-acetate) polymer (labeled E1420), was used to obtain spray-dried microparticles loaded with DSA. EN is easily dispersible in water and capable of masking off-flavors and -odors [21,22]. To evaluate the polymer’s ability to protect and release DSA and to improve the shelf life of the final products, all the produced microparticles were subjected to the aforementioned biological tests and to the technological characterization (DSC, FTIR, SEM, dissolution tests, accelerated stability studies). Finally, the results were compared with those from pure DSA. 

## 2. Results and Discussion

### 2.1. Almond Skin (AS) and Decoction Preparation: Parameters Selection

Almonds have significant water content [22,23]. Temperature (T °C) and relative humidity (RH) are essential factors influencing their shelf-life quality. The scientific literature reports that at a T °C of −3 °C/0 °C almonds have a shelf life of approximately one year; a temperature between 5 °C and 25 °C can be used, but temperatures > 30 °C improve preservation for more extended periods [23]. For this reason, a drying temperature of 40 °C ± 2 °C was chosen. With these, a reduction in %RH of approximately 86% was followed by a potential further increase in the stability of almond skins [24] (Appendix A).

The decoction and the use of 100% water solvent were chosen to produce a “green” edible extract for nutraceuticals or health food. The obtained aqueous extract (DSA) was lyophilized (DSA lio) (yield: 9.0 ± 1.0%, *w*/*w*) to improve its shelf life.

### 2.2. DSA Preliminary Chemical Characterization and Markers Choosing

A rapid HPLC-UV method was applied to investigate the DSA lio polyphenolic profile [25]. Nine unknown compounds within acid phenolic, flavanol, and flavonol classes were observed: gallic acid (1), procyanidin B3 (2), procyanidin B1 (3), catechin (4), epicatechin (5), procyanidin C1 (6), kaempferol 3-*O*-rutinoside (7), isorhamnetin-3-*O*-rutinoside (8), and quercetin (9) (Appendix A). The compounds were identified by comparing their retention time (Appendix A) and mass spectral data (Table 1 and Appendix A) with those of pure standards. Phenolic compounds reported in this study were found in the skins of several almond varieties, and they could be responsible for the extracts’ activity [3,26]. (Epi)catechin and its oligomers, known as proanthocyanidin, are almond skins’ most abundant phenolic constituents [27]. Therefore, catechin (CA) was chosen as the UV marker, while procyanidin B3 and CA were HPLC markers. The CA content was 0.8%, according to UV-Vis analysis, while the HPLC method recorded an amount of procyanidin and CA of 1.3 and 0.9 mg/g of the extract, respectively (Table 2).

### 2.3. Preliminary DSA In Vitro Biological Activities and Safety

The biological activities were carried out to evaluate the potential use of DSA in antioxidant and anti-inflammatory nutraceutical formulations for antimetabolic diseases.

#### 2.3.1. Antioxidant Assays

In recent years, almond skin polyphenols-rich extracts have been accounted as potential ingredients in designing human health products due to their high antioxidant activity [28]. Additionally, using antioxidants to contrast oxidative stress is considered an innovative strategy to prevent dysmetabolic disease [29]. The level of antioxidant capacity has been widely used to characterize antioxidant-rich foods and beverages [30]. The antioxidant action can arise from the direct scavenging of different reactive oxygen species, such as HO•, ROO•, NO•, and LOO•, and the antioxidants may respond differently to various radical or oxidant sources [31]. Multiple reaction characteristics and mechanisms are usually involved in oxidative damage, and a single assay does not accurately reflect the radical and antioxidant sources in a mixed or complex system. Considering these aspects, a set of antioxidant tests was carried out to predict the potential antioxidant capacity of DSA-rich polyphenol extract. DPPH and ABTS, the most popular methods to evaluate the antioxidant activity of foods and natural products, are rapid, simple, and inexpensive. However, their use is limited to nonphysiological radicals. On the contrary, the ORAC assay, elected as the golden standard of the pharmaceutical industry, is based on the inhibition of the oxidation of a fluorescent substrate (and fluorescence loss) by peroxyl radicals that reflect physiologically relevant perturbations [32]. Apart from that, the capability of DSA to inhibit the Nitric Oxide (NO) radical spontaneous production from a sodium nitroprusside solution was also investigated. In fact, NO is a free radical that plays a key role in normal body functions such as neurotransmission, synaptic plasticity, vasodilatation, and anticancer activities. However, a high concentration of NO is associated with pathophysiologic states like diabetes, multiple sclerosis, arthritis, carcinomas, and Alzheimer’s diseases [33].

CA was selected as a reference compound for all antioxidant tests. The results in Table 3 show that the lyophilized extract has an important antioxidant capacity in all the assays. DSA exhibited a significant (*p* < 0.05) and dose-dependent free radical scavenging activity against DPPH• and ABTS•+ radicals (SC_50_ = 211.6 µg/mL and TEAC values = 0.3 mM Trolox/mg of extract, respectively) in comparison with CA (SC_50_ = 5.9 µg/mL and TEAC values = 2.8 mM Trolox/mM of the compound, respectively). Moreover, the reduction in fluorescence of a fluorescein-AAPH solution was higher than Trolox, the positive control, and CA (ORAC units = 4.2, 1.0, and 2.0 µmol TE/µg of sample, respectively). The lyophilized extract at assayed concentration was more active in inhibiting NO production concerning the positive control, Curcumin, and CA (NO inhibition = 63.1, 42.0, and 59.0%, respectively). The antioxidant efficacy of DSA could be because of its TPC content (56.7 ± 0.9 mg GAE and 83.6 ± 1.6 mg CAE/g of extract).

#### 2.3.2. Advanced Glycation End Products (AGEs) and Metalloproteinases’ (MMPs’) Inhibition

Several studies have highlighted that AGE–RAGE interactions cause the production of ROS due to NADPH oxidase stimuli. This result was associated with dysmetabolic diseases [16,34,35]. Furthermore, the binding of AGEs to their receptors (RAGEs) stimulates various signaling pathways, such as the transcription of nuclear factor kappa B (NF-kB), stimulating the synthesis and release of proinflammatory cytokines and MMPs [9,15]. Therefore, introducing natural antioxidants such as polyphenols with AGEs’ and MMPs’ inhibitory capacities in a daily diet helps in the case of inflammatory and dysmetabolic pathologies [13,14,22]. 

Figure 1 shows the high AGEs inhibition capacity of the extract, capable of reducing the fluorescence value of the positive control (BSA with fructose, corresponding to the max AGEs fluorescent formation) from 37,042 to 13,250 nm. Instead, CA, the reference marker, shows a fluorescence value of 16,452 nm superimposable to that of the AMG (RFU = 17,474 nm), a reference compound well known for its AGEs inhibition property. The percentage of AGE inhibition was also calculated and is reported in Table 4. An impaired pattern of MMPs, characterized by overexpression and activation, is evident in subjects with metabolic syndrome [36]. In particular, gelatinases A and B (MMP-2 and -9), secreted by several vascular cell types, are implicated in vascular matrix remodeling and seem to be involved in many dysmetabolic vascular complications. The inhibitory effect of lyophilized extract and CA on MMP-2 and -9 was addressed. The results, expressed as IC_50_, showed the catechin activity only on MPP-2, while DSA had a higher inhibitory activity on MPP-2 than on MPP-9.

#### 2.3.3. DSA Safety Evaluation

To verify the extract safety, DSA was assayed in vitro with MTT, an indicator of cell viability, against a panel cell line: human malignant melanoma (A375), alveolar adenocarcinoma (A549), and epidermal keratinocyte (HaCaT). The obtained data obtained in our experimental models have shown that different concentrations (50–25–10–5–2.5 μM) of DSA raw material do not imply cell viability reduction on analyzed cell lines (Appendix A). This finding supports the safe use of the produced extract as a potential ingredient of food and nutraceutical proposals.

### 2.4. DSA and Microparticles Technological Characterization

#### 2.4.1. DSA

The in vivo absorption and bioavailability, crucial parameters of oral administration, are influenced by morphology, water solubility, and in vitro dissolution rate. Spray-dryer technology reduced the DSA crystalline structure, making it more amorphous (Figure 2) and with a greater dissolution rate (Figure 3). Unfortunately, DSA adhered to the chamber wall of the spray dryer, probably due to the sugar moieties in the extract, and the production yield was only 5%. On the other hand, the DSA powder extract remained sticky.

Solubility evaluation is a limiting factor of in vivo dissolution and absorption after oral administration. According to USP 41, DSA (DSA lio) is considered low soluble in water (2.6 g/L ± 0.4 mg/L) at room temperature. No more than 14.5% of DSA dissolved in 20 min in water (Figure 3). Instead, 38% of spray-dried DSA (DSA sd) dissolved at the same time, confirming the ability of spray-drying technology to improve the extract’s morphological and technological characteristics. DSC analysis confirmed the amorphization capacity of spray-drying technology. In DSC thermograms, the DSA crystalline peak, at 290 °C, disappears (Figure 4).

In order to improve the extract’s handling, solubility, and dissolution rate, a food-grade matrix [22], EN, was chosen to have spray-dried microsystems in 1:1 (ENDSA1) and 3:1 (ENDSA3) polymer/DSA lio ratio, improving the physicochemical aspects and technological characteristics of the extract. 

#### 2.4.2. Microparticle Characterization

IE value for all microparticles resulted in 99.98 ± 1.02 and 100.1 ± 1.01% for, respectively, 1:1 and 3:1 polymer/extract ratio. 

Despite the small amount of material used (1.32–2 g/200 mL of water), the presence of adhesive sugar moiety in the dry chamber, and the tiny microparticles aspirated during the drying process, the %Yield of all microparticles produced (about 50.0% for both types of microparticles produced) was higher than DSA sd (5.0%), and these results could be considered satisfactory.

SEM observation of ENDSA1 and ENDSA3 (Figure 2) highlights that with increasing the amount of EN, microparticle dimensions were reduced. In fact, ENDSA1 showed a dimensional range from 1.0 μm to 4.0 μm, similar to those of EN unloaded microsystems (EN sd). ENDSA3 resulted as smaller (0.5–3.0 μm), and aggregates decreased. Likely, this is related to the greater amount of EN, capable of forming a better film around DSA, ameliorating the microparticle separation during the drying process. All the obtained microparticles showed no external extract. However, there were many more aggregates of collapsed particles also in ENDSA1. Probably, it was due to the presence of a major amount of extract with hydrogen bond formation between EN and DSA on the surface of the microparticles. This condition increases the mutual contact of the micropowders and consequent microparticle aggregation.

FTIR spectra of microparticles were compared to those of DSA lio and EN sd to find the potential interactions between extract and polymer and the capacity of EN to cover DSA lio. No significant interactions between the coating polymer and the components of the almond extract were found. The functional groups of DSA (-OH stretching at 3275.52 cm^−1^; C-H stretching at 3649.17 and 2925.95 cm^−1^; aromatic fingerprinting from 2050.60 cm^−1^ to 1980.16 cm^−1^; C=O stretching vibrations from 1734.44 cm^−1^ to 1715.95 cm^−1^; C=C stretching vibrations at 1602.74 cm^−1^; C-O stretching from 1258.97 cm^−1^ to 1042.27 cm^−1^) showed signals ascribable to flavonoids. The characteristic signals of EN (bands at 2156.27 cm^−1^ and 1473.00 cm^−1^; the presence of an acrylonitrile- group; 1470.0 cm^−1^ due to symmetric bending of -CH2; signal at 1723.12 cm^−1^ and 1642.89 cm^−1^ due to carboxylic and amide groups derived by cyano- group hydrolysis; signal at 2928.01 cm^−1^ was due to C-H stretching and probable bonds scission) [22] remain unchanged in the spectra of ENDSA1 and ENDSA3. The small variations between the two formulations were likely determined by the different molar ratios. Furthermore, EN resulted as capable of totally encapsulating the extract, confirming SEM results. In both spectra of the loaded microspheres, there appeared only an aromatic signal around 1405–1408 cm^−1^, attributable to aromatic rings of polyphenols. Probably, some of these groups were on the surface of microspheres, forming plausible hydrogen bond interaction between -OH groups of DSA and EN. 

DSC analysis confirmed the FTIR data. ENDSA1 and ENDSA3 thermograms (Figure 4) showed the absence of the typical DSA peaks, indicating that EN had completely encapsulated the extract in both ratios, 1:1 and 3:1.

The dissolution/release tests showed that only 38.0% of the DSA raw dissolved in 30 min, while amounts of 53.0% and 80.0% were released/dissolved at the same time from ENDSA1 and ENDSA3, respectively, with a burst effect (Figure 3). These results were related to the nature of EN, a starch capable of attracting water, eroding, and releasing a large percentage of the active ingredient in the first step of the analysis. Then, it swelled, acting as a superdisintegrant, reducing the hydrophobic interactions between DSA particles, and improving the extract wettability and in vitro water dissolution rate.

### 2.5. Accelerate (ICH Guidelines) and Functional Stability

The DSA lio Actual Active Content (ACC) was determined after one week (T7) at 40 °C using the HPLC method. The antioxidant and biological efficacy, as well as the cytotoxicity of raw material, were investigated:-after the microencapsulation process (T0) to verify if the spray-dried process was able to keep DSA functional activity;-under harsh storage conditions (t7), as previously reported.

After one week, ACC with HPLC displayed a drastic decrease (*p* < 0.05) in CA, about 90% (from 0.9 to 0.1 mg/g), and a moderate increase (*p* < 0.05) in procyanidin B3, about 30% (from 1.3 to 1.8 mg/g). These results were likely ascribable to the thermal degradation and self-polymerization of CA [37,38]. The AAC of both microparticle batches (ENDSA1 and ENDSA3) remained quite unaltered, confirming the capacity of EN to protect DSA. 

At T0, both produced microsystems (ENDSA1 and ENDSA3) showed an antioxidant activity similar to the unprocessed extract (Table 3 and Figure 3) in all in vitro biological assays. Furthermore, none of the produced microsystems displayed a cytotoxic effect on all assayed cell lines (Appendix A). Only in A549 cells, EN polymer (25 μM) caused a slight cytotoxic effect after 24 h of treatment. This confirms that the spray-dried process was able to keep DSA functional activity.

Under harsh storage conditions (t7), the free radical scavenging activity against DPPH• and ABTS•+ radicals, ORAC value, and the NO inhibition of unprocessed and processed DSA lio and formulations was also carried out. DSA lost its antioxidant activity (*p* < 0.05), probably due to the instability of its polyphenolic constituents, in particular, catechin derivatives. Microencapsulated extract activity remained almost unchanged (Table 3). Based on previous results, the EN matrix was capable of maintaining the functional activity and improving the stability of a polyphenol-rich extract.

## 3. Materials and Methods

### 3.1. Materials

Blanched Almond Skins (BAS; var. Corrente and Tuono mixture) were supplied by “Industria Lavorazione Mandorle Calafiore Paolo dei fratelli Calafiore C. Snc”, Floridia (SR), Italy. (±)catechin (CA), (-) epicatechin (EP), gallic acid (GA), procyanidin B1, procyanidin B3, procyanidin C1, isorhamnetin 3-O-rutinoside, kaempferol-3-O-rutinoside, Folin–Ciocalteu’s phenol reagent, 2,2-diphenyl-1-picrylhydrazyl radical (DPPH), 2,2′-Azino-bis(3-ethylbenzthiazoline-6-sulfonic acid (ABTS), fluorescein (FL), 2,2′ azobis (2-methylpropionamide) dihydrochloride, 97% (AAPH), Trolox, 4-(2-Hydroxyethyl)piperazine-1-ethanesulfonic acid (HEPES), aminoguanidine bicarbonate 97% (AMG), Bovine Serum Albumin (BSA), D-(-)fructose, and sodium azide (NaN_3_) were obtained from Merck Life (Milan, Italy). OmniMMP Fluorescent Substrate Mca-Pro-Leu-Gly-Leu- Dpa-Ala-Arg-NH2, MMP-9 (refolded) (human) (recombinant) (Catalytic Domain) and MMP-2 (catalytic domain) (human) (recombinant) were purchased from Vinci-Biochem S.r.l. (Firenze, Italy). All solvents used in this study were analytical grade and were found on the market. Human malignant melanoma (A375), alveolar adenocarcinoma (A549), epidermal keratinocyte (HaCaT), and all the supplements for cell cultures were obtained from Gibco Life Technology Corp. (Thermo Fisher Scientific, Milan, Italy).

### 3.2. Methods

DSA was phytochemically (DSA yield extraction process, qualitative and quantitative analyses), biologically (antioxidant and free radical scavenging activity, safety with MTT test), and technologically (calculation of DSA marker equivalents, DSA water solubility and in vitro dissolution rate, raw materials’ morphology) evaluated to develop oral formulations safe for human health.

#### 3.2.1. Extract Preparation

##### Dried Blanched Almond Skin Preparation and Relative Humidity Control

BASs were dried at 40 °C ± 2 °C for 6 h and then stored vacuum sealed in clear plastic bags at room temperature (25 ± 5 °C). Relative humidity (%RH) was calculated on three samples using a gravimetric method. 

##### Decoction Procedure and Production Yields

100 g of homogenate BSA was boiled in 100 mL of deionized water for 10 min and cooled. PAP3 was centrifugated four times at 5000 rpm for 10 min. The supernatant was collected and lyophilized. The production yields were determined using a gravimetric method with a balance (Crystal 100 CAL, Gibertini, Milan, Italy) and expressed as % weight of the final product with respect to the weight of the unprocessed materials. 

#### 3.2.2. Chemical Characterization

##### Chemical Profile and Quantitative Analysis of DSA

The phytochemical profile of DSA was obtained using the modified HPLC method as follows [26]: a system Agilent 1260 (Agilent Technologies, Waldbronn, Germany) furnished of a quaternary pump (Model G-1312B), a degasser (Model G-4225A), a manual injector (Rheodyne Model 7725i), and a DAD detector (Model G 4212-A) was used for chromatographic separation. The amount of 3 mg of lyophilized was mixed with water/methanol (1:1) and placed in an ultrasonic bath for 15 min at 40 °C. After that, 20 μL was analyzed in a nucleodur 100-5 C_18_ EC column (150 × 4.6 mm, 5 μm) (Macherey-Nagel, Duren, Germany) using as the mobile phase a mixture of water (A) and methanol (B), both containing 0.1% (*v*/*v*) formic acid. The chromatographic conditions: DAD detector set at 280 nm; elution gradient start 10→80% B from 0 to 40 min; 80→100% B from 40 to 43; 100% B from 43 to 48 min, 100→10% B from 48 to 54 min; flow rate at 0.8 mL/min. In order to identify and integrate the peaks, the AgilentLab Advisor Basic Software 2012 was used. The unknown peaks in the sample were identified by comparing their retention times, UV, and ESIMS spectra with those of pure standards and then confirmed with co-injection. 

##### ESI-FT-ICR-MS Analysis

Mass spectra were acquired on a solariX XR 7 T FTICR-MS equipped with an electrospray ion source ESI (Bruker Daltonik GmbH, Bremen, Germany). Mass spectra were acquired in negative ion mode in a mass range of 50–1000 *m*/*z*. The capillary voltage was set to 3.9 kV, with a nebulizer gas pressure of 1.2 bar and a dry gas flow rate of 4 L/min at 200 °C. Samples (10 mg/L) in a mixture of water and methanol, 1:1) were infused at 4 µL/min, with an ion accumulation of 20 ms, with 16 scans, and using 2 million data points (2M). Before the analysis, the mass spectrometer was externally calibrated with NaTFA. High mass accuracies were reached for the NaTFA calibration data sets, with a root mean square (RMS) error lower than 0.2 ppm. MS/MS of the ion of interest was obtained by isolating in the quadrupole and manually ramping the collision energy. The FT-ICR MS mass spectra were processed using the Bruker DataAnalysis 4 software. 

##### Quantitative Analysis Using HPLC Method and Total Phenols Content (TPC)

The quantitative assessment of DSA was carried out using the HPLC-DAD method operating in the same previous condition and with a calibration curve of CA. Different concentrations of pure compound (CA) (ranging from 0.0025 to 0.20 mg/mL) were prepared and analyzed (in triplicate) using the linear least-square regression equation derived from the peak area (regression equation: y = 16550x − 17.178, r^2^ = 0.9997, y is the peak area and x the concentration). TPC was analyzed according to the Folin–Ciocalteu colorimetric method [38] and expressed as gallic and catechin equivalents (GAE and CAE mg/g of dry extract, respectively).

##### UV-Vis Method

UV-Vis assay was carried out according to ICH [Q2(R1)] guidelines and validated statistically using % Relative Standard Deviation (% RSD). CA concentration was evaluated by measuring absorbance at λmax of 280 nm in 1 mm cell (UV-Vis 1601 Shimadzu Europa, Duisburg, Germany) according to the Lambert-Beer Law (USP 41): E^1%^ _1cm_ × c × l
where E^1%^ _1cm_ is the absorbance of 1 g/100 mL (1% *w*/*v*) solution in a 1 cm cell, c is the concentration of the solution (g/100 mL), and l is the path length of the cell where the sample is held. In order to validate the method, active concentration was calculated using the standard calibration curve. *Precision.* The intraday and interday precision was determined by analyzing the same concentration of CA (10 mg/L) for, respectively, 6 h and 3 days. *Linearity.* The proportionality between absorbance and concentration was verified at room temperature at three concentration levels in the range of 2.5 to 150.0 mg/mL (y = 151.47x − 0.754, R^2^ = 0.997, where y is the absorbance, and x is the concentration used). Reference standard solutions were prepared (5 mL) and analyzed in triplicate. The results are expressed as % average value ± % RSD. 

#### 3.2.3. In Vitro Biological Activity

##### Bleaching of the Free Radical 1,1-Diphenyl-2-picrylhydrazyl (DPPH Test)

The stable radical 1,1-diphenyl-2-picrylhydrazyl was used to evaluate the free radical scavenging activity of DSA and catechin (used as positive control) following a previously described method [39]. The mean effective scavenging concentrations (SC50) of extract or control that reduced 50.0% of the initial concentration of DPPH° were calculated using the GraphPad Prism 7.0 software (San Diego, CA, USA). Low values of SC_50_ indicated a higher scavenging capacity.

##### TEAC (Trolox Equivalent Antioxidant Capacity) Assay

The ability of DSA to scavenge ABTS radical cation (2,2′-azino-bis (3-ethylbenzothiazolin)-6-sulfonicacid) was evaluated using the TEAC assay [39]. The antioxidant effect was expressed as mmol Trolox (an analog of vitamin E water-soluble used as a positive test control) equivalent (TE)/mg extract. High values of TEAC indicated a higher scavenging activity. 

##### ORAC Assay

The ORAC method, as follows [40], was used to determine the antioxidant capacities of DSA. Fluorescein (10 nM), the fluorescence probe, was selected as a reference compound attacked from peroxyl free radicals generated from an APPH (2,2′-azobis-2- methyl-propanimidamide, dihydrochloride) (100 mM) solution. To calculate the area under the curve (AUC) of tested compounds, the reaction was conducted at 37 °C, pH 7.0, until a fluorescence decay of Fluorescein solution in the presence of APPH. A Wallac 1420 Victor 96-well plate reader (Perkin Elmer, Waltham, MA, USA) with a fluorescence filter (excitation 485 nm, emission 520 nm) was used for measurement. Trolox (12.5 µM) was chosen as the control standard, phosphate buffer as blank, and catechin as a positive control. The ORAC value refers to the net protection area under the quenching curve of fluorescein in the presence of an antioxidant. The ORAC values, expressed in ORAC Units (equivalent to Trolox micromol per microgram of sample -µmol/µg), were calculated:ORAC value (μmol/µg) = K(S sample − S blank)/(S Trolox − S blank)
where K is a sample dilution factor, S is the area under the fluorescence decay curve of the sample, Trolox, or blank. S was calculated with Origin^®^7 (OriginLab Corporation, Northampton, MA, USA).

##### Nitric Oxide (NO) Radical Scavenger Assay

DSA’s nitric oxide radical scavenging capability was found according to our previous method [41]. A total of 100 μg/mL of extract, curcumin (reference compound), and catechin (positive control) were added to an aqueous solution of sodium nitroprusside (20 mM) at 25 °C for 3 h to inhibit the spontaneous NO production. The determination of NO levels was detected with Griess reagent (1% sulphanilamide, 0.1% naphthylethylenediamine dichloride (NED), and 3% phosphoric acid). Absorbance was measured at 540 nm with a spectrophotometer (Thermo Scientific Multiskan^®^ EX). The amount of NO radicals was calculated as follows:% of inhibition of NO = [A_0_ − As]/A_0_ × 100
where A0 and As were untreated and tested samples’ absorbance (nm), respectively. 

##### Antiglycation Effect

The inhibition of fluorescence produced by AGE formation through the Maillard reaction was evaluated according to Lauro et al., 2017 [12]. The results are reported in relative fluorescence units (RFU), and the percentage of inhibition concerning the positive control (BSA with fructose) is calculated as follows:% of inhibition = 1 − [(RFU sample nm − RFU positive control nm) × 100](1)

##### MMP-2 and MMP-9 Inhibitory Assay

The assay inhibition assay [9] was based on the inhibiting of the hydrolysis of the fluorescence-quenched peptide substrate Mca-Pro-Leu-Gly-Leu-Dpa-Ala-Arg-NH_2_ (Vinci Biochem s.r.l., Vinci, Italy). The results were plotted with Origin^®^7 (Origin Lab Corporation, Northampton, MA, USA) software and expressed as a concentration of inhibitors that reduced 50.0% of the MMP activity (IC_50_). 

#### 3.2.4. Cell Viability Assay

##### Cell Culture

Cell lines of human malignant melanoma (A375), alveolar adenocarcinoma (A549), and epidermal keratinocyte (HaCaT) were grown in Dulbecco’s modified Eagle’s medium containing high glucose supplemented with 10% fetal bovine serum and 100 U/mL each of penicillin and streptomycin in a humidified atmosphere of 5% CO_2_ at 37 °C. Cells were used at less than 80% of confluence.

##### Viability Assay

Cell viability was analyzed with MTT (3- [4,5-dimetiltiazol-2,5-diphenyl-2*H*-tetrazolium bromide]) assay [42] to compare the effect of the potentially cytotoxic substance with a control condition. Briefly, cells (3.5 × 10^3^/well) were grown in 96-well plates and, after 24 h, were treated with fresh medium alone or containing various concentrations (100–50–25–10–5–2.5 μM) of our compounds (DSA- ENDSA 1:1-ENDSA3:1) and catechin (100–50–25–10 μM), used as standard, and cells were incubated for 24 h. The assay was conducted on the dissolved decoction using both DMSO and water. Staurosporin 0.2 µM was used as a positive control. After the treatment, 25 μL of MTT (5 mg/mL) was added to each well, and plates were incubated for an additional 3 h, allowing salt formazan to crystallize. Subsequently, salt formazan was solubilized with 100 μL of a solution at pH 4.5, containing 50% (*v*/*v*) *N*,*N*-dimethylformamide, and 20% (*w*/*v*) SDS. Absorbance at 620 nm for each well was evaluated using a Multiskan Spectrum Thermo Electron Corporation Reader. Cell vitality was determined as
% vitality = 100 × (OD _treated_/OD _DMSO_).

#### 3.2.5. Statistical Analysis

Data evaluations and statistical analysis were reported with the commercially available software GraphPad Prism 8 (GraphPad Software Inc., San Diego, CA, USA). Results are represented as mean ± standard deviation values of at least three different experiments performed in technical triplicate. Statistical analyses were obtained thanks to the nonparametric Mann–Whitney U test. The differences were considered significant if *p* values were <0.05.

#### 3.2.6. Formulation Studies and Technological Characterization

##### Feed Preparation and Microparticle Preparation

EN, a starch derivative, was suspended in hot water (80 °C), with stirring for 24 h [27], to obtain a fine dispersion (0.5% m/V). Then, DSA was added to the polymer suspension with a 1:1 or 3:1 polymer/extract ratio and stirred for 10 min to give ENDSA1 and ENDSA3 spray-dried microparticles, respectively (Mini Buchi B-290, Flawil, Switzerland). The spray-drying parameters were Nozzle, 700 µm; inlet T, 120 °C; outlet T, 65 °C; pump, 10; flow rate, 3 mL/min; aspirator, 100. Each sample was prepared in triplicate.

##### Solubility Studies

An excess of extract (60.0 mg) was introduced into a flask containing 10 mL of water. The sample, shaken for 3 days at 25 °C [43], was filtered on a 0.45 μm filter. To calculate the amount of dissolved extract, the supernatant was examined at 280 nm (UV-Vis apparatus, 1 cm cell), and the analysis was carried out in triplicate.

##### In Vitro Dissolution Test

The tests were performed in sink conditions as follows: the amount of DSA or formulation corresponding to 0.8 g of Catechin (CA) was dissolved in 1000 mL of water in a SOTAX AT Smart Apparatus, Basel, Switzerland online with a spectrophotometer (280 nm, Lambda 25 UV-Vis spectrometer, Perkin-Elmer Instruments, Waltham, MA, USA) and USP 41 dissolution test apparatus (n.2: paddle, 100 rpm at 37 °C). All the dissolution/release tests were carried out in triplicate under “sink conditions”. In the graph, the mean values are reported (standard deviations < 5%).

##### Morphology

Raw materials and all the formulations developed were analyzed using a scanning electron microscope (SEM) (Carl Zeiss EVO MA 10, Carl Zeiss s.p.a., Milan, Italy), operating at 20 kV. Dried samples were dispersed on adhesive carbon tabs (12 mm) coated with aluminum stub and metalized with a LEICA EMSCD005 sputter coater (sputter current, 30 mA; sputter time, 135 s; thick gold layer, 200–400 Å). The particle diameters were determined from an average of at least 20 observations.

##### Spray-Drying Process Yield (SPY), Actual Active Content (AAC), and Inclusion Efficiency (IE)

10.0 mg of DSA or microparticles was dissolved directly in 10 mL of deionized water and vortexed for 60 s at 3000 rpm to determine the actual active content (AAC) using the previously reported HPLC method. AAC was expressed as the catechin (CA) and gallocatechin (GC) content percentage equivalent in 100 mg of powder. Each analysis was carried out in triplicate, and the results are expressed as mean values.

The ratio of actual extract content (AEC) to theoretical extract content (TEC) was used to calculate the IE in a freeze-dried complex according to the following equation:IE (%) = (AEC/TEC) × 100

##### Fourier-Transform Infrared Spectroscopy (FTIR)

FTIR spectra were analyzed from 2000 to 600 cm^−1^ with 256 scans and 1 cm resolution (IRAffinity-1S, Shimadzu Corporation, Kyoto, Japan, MIRacle ATR with ZnSe thin crystal).

##### Differential Scanning Calorimetry (DSC)

Thermal cycle was used to evaluate the thermal behavior of each sample (indium-calibrated Mettler Toledo DSC 822e, Columbus, OH, USA). Samples were placed in a pierced 40 μL aluminum pan and scanned (10 C/min) between 25 and 350 °C. Melting temperature (Tm) and heat of fusion (DHm) were measured.

## 4. Conclusions

This work aims to develop “green” formulations starting from an aqueous extract from Sicilian almond skins and the biological evaluation of possible anti-inflammatory activities and their causes (antioxidant/radical scavenging properties, MMPs, and AGEs’ inhibition). The results of biological studies indicated that DSA is a good candidate for formulating nutraceuticals capable of supporting antioxidants, scavenging free radicals, and inhibiting inflammatory diseases. Indeed, DSA showed higher inhibitory activity on AGEs, MPP-2, and MPP-9 than catechin used as a standard. Unfortunately, the lyophilized extract was sticky, slowly soluble in water, and underwent oxidation/degradation phenomena. To overcome these problems, spray-dried microparticles were formulated using Eudraguard^®^ Natural, a GRAS acetylated starch derivative, as a coating polymer. EN was found to be an ideal polymer to carry polyphenol-rich extracts similar to DSA. It coated all extract particles, protecting DSA from harsh conditions and improving its handling, wettability, solubility, dissolution rate, and the shelf life of the final product.

## Figures and Tables

**Figure 1 molecules-28-07913-f001:**
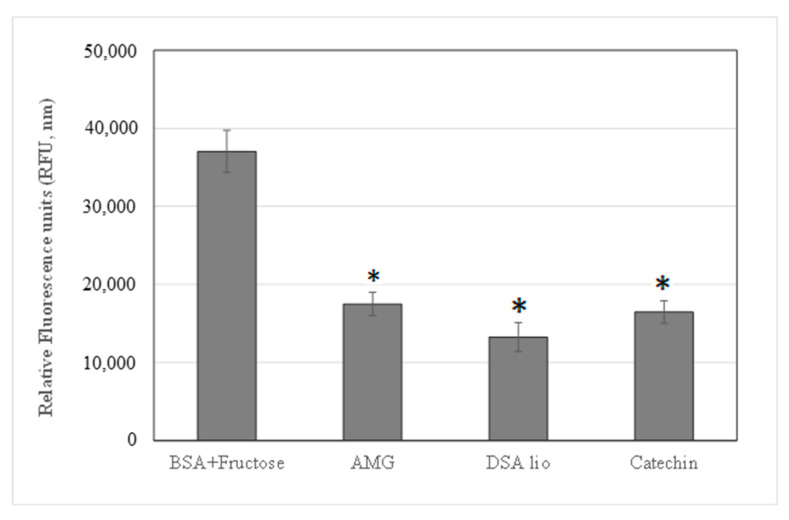
AGEs inhibition of DSA lio and catechin in comparison with AMG assay standard control. Data are expressed as mean ± standard deviation of three determinations, * *p* < 0.05 significantly different versus control (BSA with fructose).

**Figure 2 molecules-28-07913-f002:**
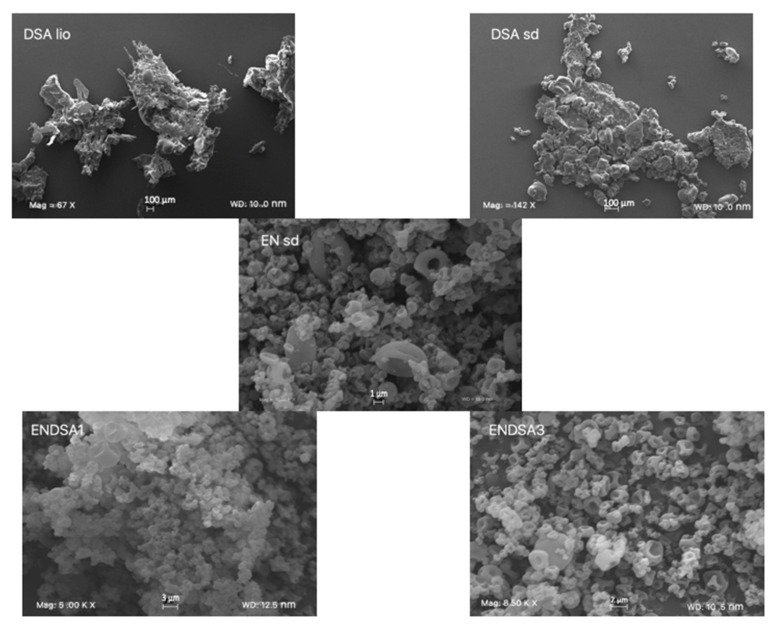
Unprocessed almond skin lyophilized extract (DSA lio), processed almond skin lyophilized extract (DSA sd), Processed Eudraguard^®^ Natural (EN sd), 1:1 EN/DSA lio microsystems (ENDSA1), and 3:1 EN/DSA lio microsystems (ENDSA3) at different magnification (Mag): DSA lio, 67 X; DSA sd, 142 X; EN sd, 10.00 KX, ENDSA1, 5.0 KX, ENDSA3, 8.5 KX. Frame average, N = 1.

**Figure 3 molecules-28-07913-f003:**
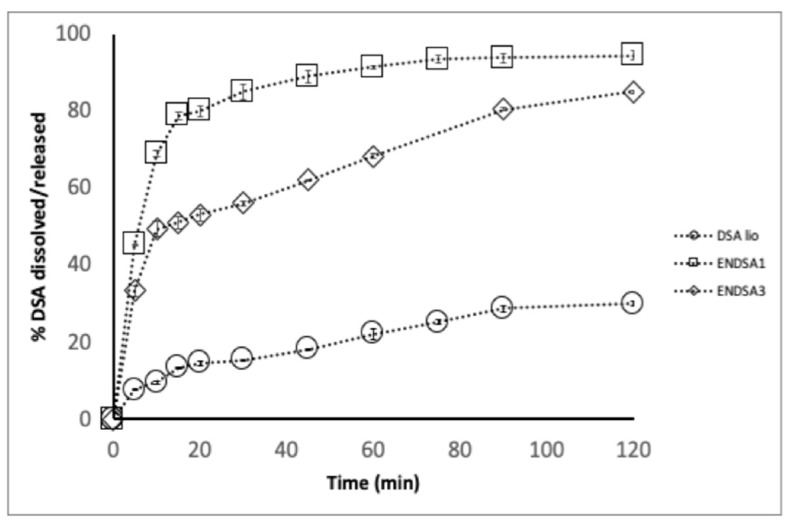
Dissolution/release profile of DSA, ENDSA1, and ENDSA3 in water.

**Figure 4 molecules-28-07913-f004:**
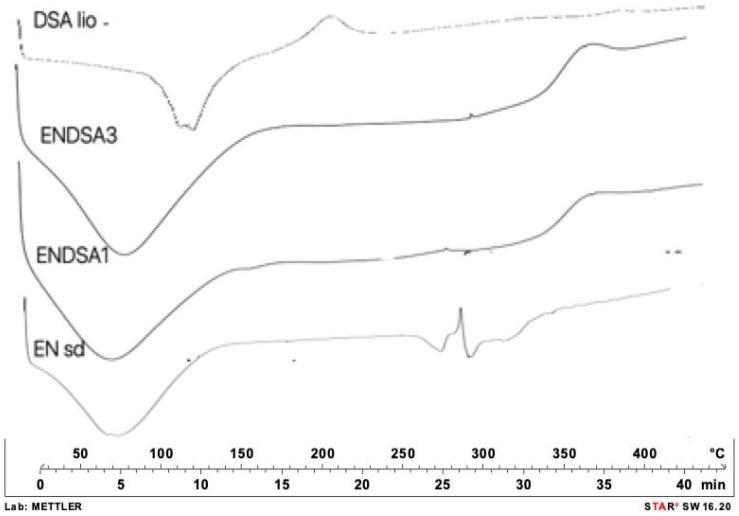
DSC thermograms of unprocessed extract (DSA lio), unloaded microparticles (EN sd), and loaded microparticles (ENDSA 1 and ENDSA3).

**Table 1 molecules-28-07913-t001:** High-resolution masses of [M − H]^−^ ions, molecular formulas, and diagnostic MS^2^ product ions of the identified polyphenols from DSA.

Compounds	[M − H]^−^ (*m*/*z*) (FT- ICR)	Mass Error (ppm)	Diagnostic MS/MS Ions (*m*/*z*)	Formula
gallic acid (**1**)	169.01436	−0.52	125.17	C_7_H_6_O_5_
procyanidin B3 (**2**)	577.1359	−1.30	425.10; 289.07	C_30_H_26_O_12_
procyanidin B1 (**3**)	577.1357	−1.0	425.10; 289.07	C_30_H_26_O_12_
catechin (**4**)	289.0722	−1.51	245.11; 205.08	C_15_H_14_O_6_
epicatechin (**5**)	289.0721	−1.38	245.11; 205.08	C_15_H_14_O_6_
procyanidin C1 (**6**)	865.1987	−0.19	577.13; 289.07	C_45_H_38_O_18_
kaempferol 3-O-rutinoside (**7**)	593.1519	−1.34	285.14	C_27_H_30_O_15_
isorhamnetin-3-O-rutinoside (**8**)	623.1618	−1.7	315.11; 300.01	C_28_H_32_O_16_
quercetin (**9**)	301.0353	−0.02	179.18; 151.18	C_15_H_10_O_7_

**Table 2 molecules-28-07913-t002:** Actual Active Content (AAC) with the HPLC-UV method of DSA lio (unprocessed extract), EDSA1 and EDSA3 (microparticles) before (T_0_) and after (T_7days_) the accelerated stability test.

	AAC ^a,b^			
	T_0_		T_7days_	
samples	procyanidin B3	catechin	procyanidin B3	catechin
DSA lio	^A,c^ 1.3 ± 0.1	^A,c^ 0.9 ± 0.1	^A,d^ 1.8 ± 0.2	^A,d^ 0.1 ± 0.01
ENDSA1	^A,c^ 1.3 ± 0.2	^A,c^ 0.8 ± 0.1	^B,c^ 1.4 ± 0.2	^B,c^ 0.7 ± 0.1
ENDSA3	^A,c^ 1.4 ± 0.3	^A,c^ 1.0 ± 0.2	^B,c^ 1.4 ± 0.5	^B,c^ 0.8 ± 0.2

^a^ Actual Active Content (AAC) was determined with the HPLC method and expressed as procyanidin B3 and catechin equivalent mg/g of extract or powder. ^b^ All results are expressed as the means ± standard deviation of three experiments performed in triplicate. Means with different small letters (c,d) within a row are significantly different (*p* < 0.05). Means with different capital letters (A,B) within a column are significantly different (*p* < 0.05).

**Table 3 molecules-28-07913-t003:** Free radical scavenging activity (DPPH assay), Trolox equivalent antioxidant capacity (TEAC assay), ORAC assay, nitric oxide radical scavenger (NO assay) of unprocessed DSA extract (DSA lio) and ENDSA1, and ENDSA3-loaded microsystems.

Samples	DPPH Assay (SC_50_ = μg/mL) ^a^		TEAC (mM TE/mg Extract or mM Compound) ^b^		ORAC (ORAC Units = μmol TE/μg Sample) ^c^		NO Scavenger (% Inhibition of NO)	
	T_0_	T_7_	T_0_	T_7_	T_0_	T_7_	T_0_	T_7_
DSA lio	^A,e^ 211,6 ± 1,9	^A,f^ 232.8 ± 1.3	^A,e^ 0.27 ± 0.01	^A,f^ 0.24 ± 0.02	^A,e^ 4.23 ± 0.20	^A,f^ 3.50 ± 0.50	^A,e^ 63.12 ± 0.19	^A,f^ 56.51 ± 1.10
ENDSA1	^B,e^ 203.2 ± 1.7	^B,e^ 207.3 ± 2.3	^A,e^ 0.30 ± 0.04	^B,e^ 0.29 ± 0.05	^A,e^ 4.26 ± 0.41	^B,e^ 4.20 ± 0.35	^A,e^ 65.10 ± 0.20	^A,e^ 63.81 ± 0.60
ENDSA3	^B,e^ 202.0 ± 2.1	^B,e^ 206.8 ± 2.1	^A,e^ 0.28 ± 0.02	^B,e^ 0.27 ± 0.03	^A,e^ 4.25 ± 0.33	^B,e^ 4.19 ± 0.25	^A,e^ 64.01 ± 0.31	^A,e^ 62.8 ± 0.24
CA	5.9 ± 0.9		2.8 ± 0.50		2.00 ± 0.50		59.00 ± 0.42	
α-tocopherol ^d^	10.1 ± 1.2							
BHT ^d^			0.36 ± 0.03					
Trolox ^d^					1.00 ± 0.07			
Curcumina ^d^							42 ± 0.28	

Data were expressed as the mean ± standard deviation of three determinations performed in triplicate; ^a^ SC_50_ ± standard deviation, expressed as a mg unit of extract untreated or treated/mL; ^b^ TEAC value ± standard deviation, expressed as mM of Trolox equivalent/mg of extract untreated or treated or mM of pure compound; ^c^ ORAC units ± standard deviation, expressed as mmol of Trolox equivalent/mg of extract untreated or treated or pure compound. ^d^ a-tocopherol, BHT (Butyl Hydroxy Toluene), Trolox, and Curcumin were used as positive controls of DPPH, TEAC, ORAC, and NO assays, respectively. Means with different small letters (e,f) within a row were significantly different (*p* < 0.05). Means with different capital letters (A,B) within a column were significantly different (*p* < 0.05).

**Table 4 molecules-28-07913-t004:** Antiglycation effect and MMPs’ inhibition of lyophilized extract (DSA lio), in comparison with catechin (CA).

Samples	AGEs (% Inhibition)	MMP-2 (IC_50_ µg/mL)	MMP-9 (IC_50_ µg/mL)
DSA lio	64.23 ± 0.23	8.10 ± 0.17	45.23 ± 0.35
CA	55.59 ± 0.45	22.23 ± 0.26	n.a.
AMG	56.83 ± 0.242		

Data are expressed as mean ± standard deviation of three determinations; n.a. = no active; AMG (aminoguanidine): reference compound.

## Data Availability

Data are contained within the article.

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
