# Peer review of "A Green Bioactive By-Product Almond Skin Functional Extract for Developing Nutraceutical Formulations with Potential Antimetabolic Activity"

_molecules, 2023, doi:10.3390/molecules28237913_

Round 1

Reviewer 1 Report

Comments and Suggestions for Authors

Almond peel extract showed good antioxidant and free radical activity with a stronger NO inhibition effect, strong activity on MMP-2, and good antiglycative effects. In light of this, a food supplement with added health value has been formulated. Eudraguard® Natural acted as a swelling substrate by improving extract solubility and dissolution/release. Almond peel extract was an optimal candidate to formulate safe microsystems with possible anti-metabolic activity.

There is room to optimize the framework and logic of this manuscript as well as the diagrams. There still have some space to improve for this manuscript. Other questions were shown below:

1.      The title of the manuscript is too vague and does not reflect the specific aim and scope of the study.

2.      Line 51-53. “Generally, anti-inflammatory activity is due to several mechanisms such as antioxidative and radical scavengers’ properties [7-8] of polyphenol compounds, with also direct or indirect MMPs inhibition and AGEs reduction [8].” There is positive correlation about polyphenols and antioxidative effect, please refer this reference (Food Chemistry, 402(2023): 134231). While the polyphenols inhibit the AGEs reduction, please refer this reference (Food Bioscience. 43(2021), 101313. Critical Reviews in Food Science and Nutrition. Doi: 10.1080/10408398.2022.2076064).

3.      Table 1. Table 2. It should be compared with different vibration.

4.      Is there any correlation about antioxidant activity and AGEs? (Food Chemistry, 417(2023):135861).

5.      References should be in a standard format, with appropriate references to the latest literature.

Comments on the Quality of English Language

Almond peel extract showed good antioxidant and free radical activity with a stronger NO inhibition effect, strong activity on MMP-2, and good antiglycative effects. In light of this, a food supplement with added health value has been formulated. Eudraguard® Natural acted as a swelling substrate by improving extract solubility and dissolution/release. Almond peel extract was an optimal candidate to formulate safe microsystems with possible anti-metabolic activity.

There is room to optimize the framework and logic of this manuscript as well as the diagrams. There still have some space to improve for this manuscript. Other questions were shown below:

1.      The title of the manuscript is too vague and does not reflect the specific aim and scope of the study.

2.      Line 51-53. “Generally, anti-inflammatory activity is due to several mechanisms such as antioxidative and radical scavengers’ properties [7-8] of polyphenol compounds, with also direct or indirect MMPs inhibition and AGEs reduction [8].” There is positive correlation about polyphenols and antioxidative effect, please refer this reference (Food Chemistry, 402(2023): 134231). While the polyphenols inhibit the AGEs reduction, please refer this reference (Food Bioscience. 43(2021), 101313. Critical Reviews in Food Science and Nutrition. Doi: 10.1080/10408398.2022.2076064).

3.      Table 1. Table 2. It should be compared with different vibration.

4.      Is there any correlation about antioxidant activity and AGEs? (Food Chemistry, 417(2023):135861).

5.      References should be in a standard format, with appropriate references to the latest literature.

Author Response

Reviewer 1

Almond peel extract showed good antioxidant and free radical activity with a stronger NO inhibition effect, strong activity on MMP-2, and good antiglycative effects. In light of this, a food supplement with added health value has been formulated. Eudraguard® Natural acted as a swelling substrate by improving extract solubility and dissolution/release. Almond peel extract was an optimal candidate to formulate safe microsystems with possible anti-metabolic activity.

Dear Reviewer 1,

Thank you for your comments. We revised the manuscript, improving the quality and the English language according to your suggestions. All modified parts have been highlighted in yellow.

There is room to optimize the framework and logic of this manuscript as well as the diagrams. There still have some space to improve for this manuscript. Other questions were shown below:

  1. The title of the manuscript is too vague and does not reflect the specific aim and scope of the study.

Thank you for your suggestion. As requested, the title has been modified to” A green bioactive by-product almond skins functional extract to develop nutraceutical formulations with potential anti-metabolic activity.”

  1. Line 51-53. “Generally, anti-inflammatory activity is due to several mechanisms such as antioxidative and radical scavengers’ properties [7-8] of polyphenol compounds, with also direct or indirect MMPs inhibition and AGEs reduction [8].” There is positive correlation about polyphenols and antioxidative effect, please refer this reference (Food Chemistry, 402(2023): 134231). While the polyphenols inhibit the AGEs reduction, please refer this reference (Food Bioscience. 43(2021), 101313. Critical Reviews in Food Science and Nutrition. Doi: 10.1080/10408398.2022.2076064).

The references have been amplified, and those suggested have been inserted into the list (references n.10,13,14). The text has been modified to:

"These effects are mainly due to the polyphenol’s antioxidant and radical scavenging properties [8-10]. Indeed, reactive oxygen (ROS) and nitrogen (RNS) species are factors strictly related to inflammation. In the presence of oxidative stress, the glycation end products (AGEs) accumulated within body tissues contribute to raising the levels of pro-inflammatory factors, causing the up-regulating of the matrix metalloproteinases (MMPs) production [9, 11-12]. Polyphenols can act on various inflammatory disease factors, inhibiting different steps of the glycation process and the MMPs' activities [9, 13 -16]. Polyphenols with multiple hydroxyl groups, such as catechins, can react with ROS and RNS in a termination reaction, breaking the new radical generation cycle [17] and capturing α-dicarbonyl species responsible for forming mono- and di-adducts. So, they inhibit the formation of AGEs [15]."
  1. Table 1. Table 2. It should be compared with different vibration.

As suggested, Table 1 and Table 2 have been changed. Different capital or small letters have been inserted to understand better the statistically significant results (p< 0.05).

  1. Is there any correlation about antioxidant activity and AGEs? (Food Chemistry, 417(2023):135861).

 Yes, there is a correlation between antioxidant activity and AGEs. The suggested reference has been inserted in the text (reference n. 15), and the correlation between antioxidant activity and AGEs has been clarified:

Lines 57-66: "Indeed, reactive oxygen (ROS) and nitrogen (RNS) species are factors strictly related to inflammation. In the presence of oxidative stress, the glycation end products (AGEs) accumulated within body tissues contribute to raising the levels of pro-inflammatory factors, causing the up-regulating of the matrix metalloproteinases (MMPs) production [9, 11-12]. Polyphenols can act on various inflammatory disease factors, inhibiting different steps of the glycation process and the MMPs' activities [9, 13 -16]. Polyphenols with multiple hydroxyl groups, such as catechins, can react with ROS and RNS in a termination reaction, breaking the new radical generation cycle [17] and capturing α-dicarbonyl species responsible for forming mono- and di-adducts. So, they inhibit the formation of AGEs [15]."

Lines 162-167: "Several studies have highlighted that AGE-RAGE interactions cause the production of ROS due to NADPH oxidase stimuli. This result has been associated with dysmetabolic diseases [16, 34-35].
Furthermore, the binding of AGEs to their receptor (RAGE) stimulates various signaling pathways, such as the transcription of nuclear factor kappa B (NF-kB), stimulating the synthesis and release of proinflammatory cytokines and MMPs [9,15]."

  1. References should be in a standard format, with appropriate references to the latest literature.

References have been revised according to the “Molecules” guide to authors, and the latest literature has been added.

A native English speaker has revised the English language.

Reviewer 2 Report

Comments and Suggestions for Authors

Shouldn't "valorized" in the title be "valorize" instead?

A comma "." should not be present in the title.

The conclusion should serve as a concise summary of the study's key findings and their implications, highlighting its significance and potentially suggesting avenues for future investigations. In the abstract, the Conclusion seems very general, not specific, so please revise. 

The English language is not good, it needs fixing in order for the study to be understandable.

There are multiple strikeouts along the text.

Some parts of the manuscript need to be placed in a new line, separate distinct sections of your paper for clarity and organization, please. Like in line 53, you end with the literature survey, and "In the present study, .." should be in a new line.

Line 42, "Skins ..." sentence is not in a grammatically correct English.

Line 45, very unclear sentences.

Line 49, "acetonic extract" of what?

Line 83 and 84 duplicate.

For the identified substances, you should present their MS/MS spectra in the supplementary file, and in the manuscript, you should prepare a LC-MS table, stating their elemental composition, molecular ion, fragment ions with their intensities, and the ppm error.

Comments on the Quality of English Language

English language is not good. Sentences are very unclear. Needs reformatting from a native English speaker.

Author Response

Dear Reviewer 2,

Thank you for your comments. We revised the manuscript, improving the quality and the English language according to your suggestions. All modified parts have been highlighted in yellow.

Shouldn't "valorized" in the title be "valorize" instead?

A comma "." should not be present in the title.

We are sorry for the typos.

The title has been modified as suggested by Reviewer 1 as follows:” A green bioactive by-product almond skins functional extract to develop nutraceutical formulations with potential anti-metabolic activity.”

The conclusion should serve as a concise summary of the study's key findings and their implications, highlighting its significance and potentially suggesting avenues for future investigations. In the abstract, the Conclusion seems very general, not specific, so please revise.

The abstract and manuscript conclusions have been revised and modified as suggested to: "

almond peels extract has a significant antioxidant activity and MMPs/AGEs inhibition effects, resulting in an optimal candidate to formulate safe microsystems with potential anti-metabolic activity. Eudraguard® Natural is capable of getting spray-dried microsystems with an improvement of the extract‘s biological and technological characteristics. It also protects the dry extract from degradation and oxidation, prolonging the shelf-life of the final product.

The English language is not good, it needs fixing in order for the study to be understandable.

There are multiple strikeouts along the text.

A native English speaker has revised the English language.

Some parts of the manuscript need to be placed in a new line, separate distinct sections of your paper for clarity and organization, please. Like in line 53, you end with the literature survey, and "In the present study." should be in a new line.

Thank you for your attention. Various parts have been placed in new lines as suggested.

Line 42, "Skins ..." sentence is not in a grammatically correct English.

The sentence has been revised as follows: “In particular, almond skins are potential source of bioactive components like vitamins, free amino acids, minerals, and polyphenols such as flavanols, primarily catechin and epicatechin (total monomer concentration of 7.8 mg/100 g). “

Line 45, very unclear sentences.

The sentences have been modified: “Once introduced to the body, catechins block free radicals, preventing DNA and protein damage. They also reduce inflammation, triggering factor in metabolic syndrome, and decrease the degradation of cholesterol plaques, minimizing the incidence of cardiovascular diseases and tumors. [4-5]. In the literature, some almond skin extracts showed activities similar to those of catechins. They have hepato-protective properties due to inhibiting hepatocyte lipid peroxidation and cytotoxicity [6].

Line 49, "acetonic extract" of what?

Yes, it has an acetonic extract. We apologize for the omission and added the word “acetonic” in the text.

Line 83 and 84 duplicate.

Thank you for your attention. The duplicate has been deleted.

For the identified substances, you should present their MS/MS spectra in the supplementary file, and in the manuscript, you should prepare a LC-MS table, stating their elemental composition, molecular ion, fragment ions with their intensities, and the ppm error.

The authors indicate that ESIMS and not LC-MS have been performed. The mass spectra have been added in Figure S2 of supplementary materials. As reported in Materials and Methods, mass data were obtained by directly infusing ESI-MS/MS of peaks collected at HPLC-DAD. In addition, an ESIMS table stating their elemental composition, molecular ion, fragment ions with their intensities, and the ppm error was added to the text (Table 1).

Comments on the Quality of English Language

English language is not good. Sentences are very unclear. Needs reformatting from a native English speaker.

As suggested, the English language has been revised by a native English speaker.

Reviewer 3 Report

Comments and Suggestions for Authors

Almond peels are rich in bioactive ingredients such as vitamins, free amino acids, minerals and polyphenols such as catechin and epicatechin. Almond peel extracts have shown, similarly to catechins, hepatoprotective properties due to the inhibition of hepatocyte lipid peroxidation and cytotoxicity. In this study, the green decoction method was used to obtain an aqueous extract from almond skins (DSA). Then, the possible anti-inflammatory effect of almond peels and their use in nutraceutical preparations were assessed. Nutraceutical recipes have also been developed in the presence of the natural polymer Eudraguard®, gluten-free and certified GMO-free, based on corn starch (starch acetate).

To assess the ability of the polymer to protect and release DSA and improve the durability of the final products, biological tests and physicochemical characterization were performed, namely DSC, FTIR, SEM, dissolution tests, accelerated stability tests.

The results of biological studies indicated that DSA is a good candidate for the formulation of nutraceuticals capable of supporting antioxidants, scavenging free radicals and inhibiting inflammatory diseases.

The novelty of this work lies in conducting a series of both biological and physicochemical tests to assess the potential health properties of almond peel extract. This information will be important when using almonds in the preparation of medical or health foods as potential treatments for specific medical conditions. The work in its current form is eligible for publication.

Author Response

Dear Reviewer 3,

Thank you very much. We are grateful for your opinion and positive comments on our manuscript.

We revised the manuscript, improving the quality and the English language. All modified parts have been highlighted in yellow.

Reviewer 4 Report

Comments and Suggestions for Authors

Comments

Introduction
Line 53, typo "also direct or indirect MMPs inhibition".
Line 54. typo "a decoction green method was applied to obtained".
Line 61, typo "DDPH", it's DPPH.
Line 61 and 62, the chemical formula of ABTS is incorrect. The correct is 2,2'-Azino-bis(3-ethylbenzothiazoline-6-sulfonic acid).

Materials and methods

Please add the number for each subtitle, example 3.1. Materials. 3.1., Methods, Extract Preparation, etc. Please revise the Author Guidelines of the journal.

Some names of chemicals are capital letters (Procyanidin B1, B3, C1, Sodium azide, etc.), why?

Line 310, the chemical formula of ABTS is incorrect, please correct it.

Symbol of degrees Celsius most be separated from the number “dried at 40 °C ±2 °C for 6 h”, also the symbol for hours is “h”. Use the correct unit, revise the Author Guidelines of the journal about it.

Line 303, 100 g.

Line 345, Successively, 20 mL.

Line 349, 280 nm.

Line 373, Gallic is in capital letters, why?

Line 385, for 6 hrs. Use the correct symbol for hours (h) along the manuscript.

Line 388, mgL−1

Author Response

Dear Reviewer 4,

Thank you for your comments and suggestions. We revised the manuscript, improving the quality and the English language. All modified parts have been highlighted in yellow.

Introduction
Line 53, typo "also direct or indirect MMPs inhibition".
Line 54. typo "a decoction green method was applied to obtained".
Line 61, typo "DDPH", it's DPPH.
Thank you for your attention. We are sorry for the typos. All typos have been corrected.

Line 61 and 62, the chemical formula of ABTS is incorrect. The correct is 2,2'-Azino-bis(3-ethylbenzothiazoline-6-sulfonic acid).

Line 310, the chemical formula of ABTS is incorrect, please correct it.

We are sorry for the mistakes. The ABTS chemical formula was correct in all text.

Materials and methods

Please add the number for each subtitle, example 3.1. Materials. 3.1., Methods, Extract Preparation, etc. Please revise the Author Guidelines of the journal.

Thank you for your suggestion. We have been using the old Molecule template before October 31st. However, we have added the number for each subtitle as suggested.

Some names of chemicals are capital letters (Procyanidin B1, B3, C1, Sodium azide, etc.), why?

Line 373, Gallic is in capital letters, why?

We are sorry for the mistakes. All capital letters have been modified

Symbol of degrees Celsius most be separated from the number “dried at 40 °C ±2 °C for 6 h”, also the symbol for hours is “h”. Use the correct unit, revise the Author Guidelines of the journal about it.

The correct symbol "h" has been inserted and the Celsius degree symbol has been modified in the text.

Line 385, for 6 hrs. Use the correct symbol for hours (h) along the manuscript.

The correct symbol "h" has been inserted.

Line 303, 100 g.

The unit “g” has been separated from “100”

Line 345, Successively, 20 mL.

The correct unit has been 20 mL and it has been modified.

Line 349, 280 nm.

 “nm” has been separated from “280”

Line 388, mgL−1

“mg/L” has been modified in “mgL−1

Round 2

Reviewer 1 Report

Comments and Suggestions for Authors

It can be accepted in the current revision.

Comments on the Quality of English Language

It can be accepted in the current revision.

Reviewer 2 Report

Comments and Suggestions for Authors

Overall, the manuscript has been improved according to the remarks.

There are several typos: Like line 51.

It is now suitable for publication.

Reviewer 4 Report

Comments and Suggestions for Authors

Comments

Results and Discussion

Line 102, “a drying temperature of 40°C±2 °C”. Symbol of degrees Celsius most be separated from the number. Please verify it along the manuscript.